# Assessment of the sublingual microcirculation with the GlycoCheck system: Reproducibility and examination conditions

Mie Klessen Eickhoff[1], Signe Abitz Winther[1], Tine Willum Hansen[1], Lars Jorge Diaz[1], Frederik Persson[1], Peter Rossing[1,2], Marie Frimodt-Møller[1] *

1 Steno Diabetes Center Copenhagen, Gentofte, Denmark, 2 Department of Clinical Medicine, University of Copenhagen, Copenhagen, Denmark

* Marie.frimodt-moeller@regionh.dk

## Abstract

### Background

The glycocalyx is an extracellular layer lining the lumen of the vascular endothelium, protecting the endothelium from shear stress and atherosclerosis and contributes to coagulation, immune response and microvascular perfusion. The GlycoCheck system estimates glycocalyx' thickness in vessels under the tongue from perfused boundary region (PBR) and microvascular perfusion (red blood cell (RBC) filling) via a camera and dedicated software.

### Objectives

Evaluating reproducibility and influence of examination conditions on measurements with the GlycoCheck system.

### Methods

Open, randomised, controlled study including 42 healthy smokers investigating day-to-day, side-of-tongue, inter-investigator variance, intraclass-correlation (ICC) and influence of examination conditions at intervals from 0–180 minutes on PBR and RBC filling.

### Results

Mean (SD) age was 24.9 (6.1) years, 52% were male. There was no significant intra- or inter-investigator variation for PBR or RBC filling nor for PBR for side-of-tongue. A small day-to-day variance was found for PBR (0.012µm, p = 0.007) and RBC filling (0.003%, p = 0.005) and side-of-tongue, RBC filling (0.025%, p = 0.009). ICC was modest but highly improved by increasing measurements. Small significant influence of cigarette smoking (from 40–180 minutes), high calorie meal intake and coffee consumption was found. The latter two peaking immediately and tapering off but remained significant up to 180 minutes, highest PBR changes for the three being 0.042µm (p<0.05), 0.183µm (p<0.001) and 0.160µm (p<0.05) respectively.

**Data Availability Statement:** All relevant data are within the manuscript and its Supporting information files.

**Funding:** This work was funded by a grant received by PR from the Novo Nordisk Foundation, grant no. NNF 14OC0013659. The funders had no role in study design, data collection and analysis, decision to publish, or preparation of the manuscript.

**Competing interests:** I have read the journal's policy and the authors of this manuscript have the following competing interests: MKE has served as educator for Astra Zeneca (all honoraria's to institution) FP has served as consultant, on advisory boards or as educator for Astra Zeneca, Novo Nordisk, Sanofi, Mundipharma, MSD, Boehringer Ingelheim, Novartis, Amgen and has received research grants to institution from Novo Nordisk, Amgen and Astra Zeneca. PR has served as consultant, on advisory boards or as educator for Astra Zeneca, Astellas, AbbVie, Novo Nordisk, Boehringer Ingelheim, Eli Lilly, Merck, Bayer (all honoraria's to institution), has shares in Novo Nordisk and has received research grants to institution from Novo Nordisk and Astra Zeneca. This does not alter our adherence to PLOS ONE policies on sharing data and materials.

## Conclusions

Measurements with the GlycoCheck system have a moderate reproducibility, but highly increases with multiple measurements and a small day-to-day variability. Smoking, meal and coffee intake had effects up to 180 minutes, abstinence is recommended at least 180 minutes before GlycoCheck measurements. Future studies should standardise conditions during measurements.

## Introduction

The role of endothelial dysfunction in several diseases is associated to cardiovascular disease [1, 2]. The extracellular matrix covering the luminal surface of the endothelial cells is termed the glycocalyx and loss of this layer has been shown to coincide with endothelial dysfunction [3]. The glycocalyx consists of proteoglycans, glycoproteins and glycolipids and the width of the glycocalyx increases with the diameter of the blood vessel [4–6] with a continuous breakdown and repair [7]. The glycocalyx protects the endothelium from friction against the blood elements and plays an important role in the process of coagulation, lipid handling and immune response [3, 8–10].

The glycocalyx consists of a luminal layer which allows penetration of the blood cells termed the perfused boundary region (PBR) whereas the outer layer is impermeable to the blood cells protecting the endothelium from direct contact with those (Fig 1). The thickness of the PBR is shown to be a sensitive measure of the state of the capillaries and a higher PBR is related to presence of early stages of atherosclerosis, albuminuria and diabetic complications [11–13]. Furthermore, the glycocalyx contributes to the regulation of microvascular perfusion and an inverse relationship between PBR and perfusion has been demonstrated [14].

It has been a challenge to measure the size of the glycocalyx where measurements primarily have been performed invasively. However, nowadays this is possible with the use of different methods [15–17]. A new non-invasive method has been developed named the GlycoCheck system—a handheld non-invasive camera (Capiscope handheld, KK Research technology Ltd) that is placed under the tongue to obtain in vivo video recordings of the blood flow in the capillaries coupled to the GlycoCheck software.

The reproducibility of microcirculation measurements of the glycocalyx under the tongue has previously been evaluated [13, 18–22]. This was done by use of orthogonal polarization

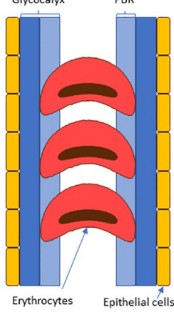
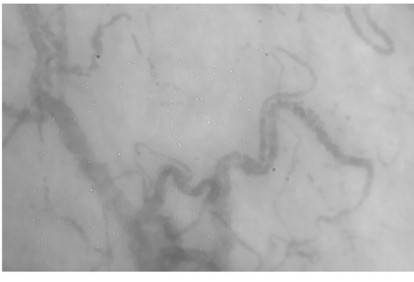

**Fig 1. Schematic representation of the lateral movement of erythrocytes and image of measured capillaries.**

spectroscopy (OPS) in some of the studies [13, 18] and by sidestream dark field (SDF) [19–22] in different patient groups.

Furthermore, there has only been one study evaluating the influence of standardised examination conditions for application of the GlycoCheck system however this was only measured at one timepoint after the exposure [19]. We therefore evaluate both intra- and inter reproducibility as well as day-to-day reproducibility in a group of healthy individuals. Furthermore, the impact of examination conditions in terms of the influence of smoking, side of the tongue measured upon, coffee consumption and intake of a high-calorie meal with repeated measurements over a period of three hours was assessed.

## Materials and methods

The GlycoCheck system has previously been described in detail [14]. In short, the system consists of a video camera attached to a computer with the GlycoCheck software. The camera emits light which is reflected on the erythrocytes and recorded by SDF imaging. The software automatically detects valid blood vessel segments for recordings and performs measurements of PBR in segments automatically positioned every 10 μm along the vessel. The camera records a sequence of 40 frames in each recording with an average of 300 valid vessel segments. A total of 3000 vascular segments and thus 10 recordings are needed for each calculation. The radial and longitudinal movements and distribution of the erythrocytes in the vessel segments are imaged and through the GlycoCheck software the thickness of the glycocalyx is calculated. This is achieved indirectly using the change in width of the erythrocytes (Fig 1) as an indicator of the extent of the lateral movement of the erythrocytes into the glycocalyx. Thus, a higher PBR indicates a thinner and more vulnerable glycocalyx. The extent to which erythrocytes are present in all 40 frames taken in one recording is an indicator of vascular perfusion and is termed the red blood cell (RBC) filling, a high value being beneficial [14].

Investigators were instructed in handling the GlycoCheck camera and system by the manufacturer. This included an instruction in avoiding using excess pressure to avoid pressure artefacts and in moving the camera to assess more than one spots in order to counterbalance for spatial heterogeneity.

### Design

Unmasked randomised controlled study on variations of measurements with the GlycoCheck system. The PBR in the capillary range from 5 to 25 μm (PBR) was our primary variable and the RBC filling the secondary variable.

### Participants

Participants were recruited by advertising at www.forsogsperson.dk, an official Danish website for recruitment of healthy participants for research studies.

Eligible were healthy smokers between 18 and 75 years of age and a total of 42 participants were included. Being healthy was defined as not receiving any medication and being a smoker was defined as smoking on average ≥ 10 cigarettes a day through a minimum of one year. All participants gave written informed consent and no study related procedures were performed beforehand.

The study was approved by the Ethics Committee of the Danish Capital Region and was conducted according to the declaration of Helsinki. Protocol nr. H-15010254.

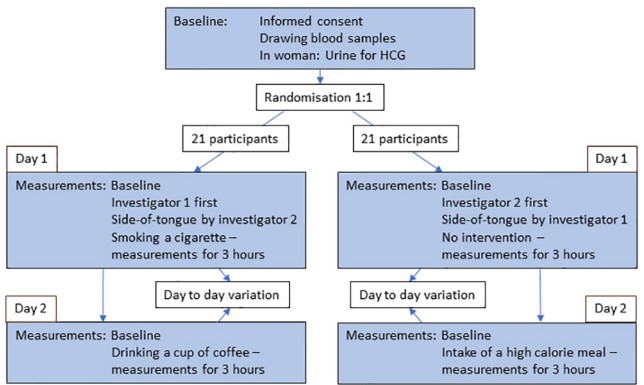

**Fig 2. Study flow chart of the examination of glycocalyx in healthy controls.**

## Study protocol

The study design is shown in Fig 2. Days 1 and 2 were no more than 7 days apart and participants had to abstain from food, beverages (except water) and smoking for eight hours before each visit day. Additionally, alcohol had to be avoided for 12 hours prior to the visits. No excessive physical exercise was allowed within three hours prior to the examinations.

The blood pressure measurements and GlycoCheck examinations were performed after 15 minutes of rest in sitting position in a quiet room at a constant temperature of 22–25 degrees Celsius. The blood pressure was measured with a validated oscillometric monitor (A&D Medical um-201, Tokyo) prior to the GlycoCheck examinations. The GlycoCheck examinations were performed with five consecutive measurements at each measuring point on the left side of the tongue, except from the side-of-tongue study where both left and right side were examined. Aside from the inter-observer variation assessment, all measurements were performed by investigator 1.

For the analysis of the effect of intervention, the measurement before intervention (baseline) was defined as the measurement performed by investigator 1 on the left side of the tongue. *Day-to-day* reproducibility was evaluated using the measurement on day 1 performed by investigator 1 on the right side of the tongue compared to the measurement before intervention on day 2.

**Day 1: Interobserver variation, left versus right side of the tongue, no intervention and smoking a cigarette.** After drawing blood samples, we first had a measuring point before intervention by either investigator 1 or 2 according to the randomisation. Then the opposite investigator had a measuring point at the left and right side of the tongue. Next the participant according to the randomisation either had no intervention or smoked a cigarette of the brand "House of Prince" containing 0,9 mg nicotine within less than six minutes. Right thereafter both groups had measurements performed at time 0 minutes. Consecutive measurements were performed at times 20, 40, 60, 90, 120, 150 and 180 minutes.

**Day 2: High-calorie meal and coffee consumption.** First a measurement before intervention was performed. Thereafter the participant was given either a high calorie meal containing 4185 kJ including 54 g fat or a cup of coffee of the brand Nescafe Original containing 75 mg caffeine. Right thereafter a measuring point was performed at time 0 minutes. Consecutive measuring points were performed at times 30, 60, 90, 120, 150 and 180 minutes.

## Statistical analyses

All characteristics are presented as means with standard deviation (SD) or if skewed distributions, as medians with interquartile range (IQR).

Sample size calculation was based on results from a study by Mulders et al. [23, 24]. A difference of 0.2 μm in PBR was suggested to be of clinical relevance with a variance of 0.3 μm, showing that at least 21 participants should be included in each group with a significance level of 0.05 and 80% power. The power calculation was confirmed using the power statement implemented in the SAS software (SAS Institute, Cary, NC), version 9.3.

The reproducibility was evaluated with the following methods: linear mixed-effects models, Bland-Altman's plots and calculation of the intraclass-correlation (ICC, as implemented in the SAS software, (3,к) where к was 3 and 5 for means of 3 and 5 measurements respectively) [25, 26]. To account for the correlation of repeated measurements within participants, statistical analysis was performed using linear mixed-effects models with a participant-specific random intercept. Bland-Altman's plots were used for limits of agreement where the differences between the studied parameters were plotted against their mean values [27]. Limits of agreement were considered to be within two standard deviations (SD) of the mean differences. Reported is both a mean of the three first and of the five first measurements in each recording round.

ICC was calculated to describe the correlation between measurements in the same individual [28] including the following in the analysis: Measuring timepoint, day, left vs. right side of the tongue, intervention and sex. We calculated ICC for means of 3 and 5 measurements respectively by two different methods: First not corrected for increasing number of measurements and secondly with the increasing number of measurements included in the analysis. An ICC equal to or higher than 0.75 was considered good, whereas 0.5 to 0.75 was considered moderate [25].

Statistical significance is inferred as a two-tailed P-value < 0.05. All analyses were performed in SAS enterprise 7.1 or R.

## Results

### Clinical characteristics

Clinical characteristics of the population are shown in Table 1; the mean (SD) age was 24.9 (6.1) years and 52% were male. First baseline measurement (mean of 5) showed a PBR of 1.995 (0.179) μm and RBC filling of 0.734 (0.043) μm. Median (IQR) days between visit 1 and 2 was 3.0 (2.0–6.0).

### Measurements

Due to technical issues (one of the four diodes in the camera was broken) 758 out of 3990 measurements were not valid due to poor video quality as visually judged by a technician from the GlycoCheck company blinded to all other data in the study and the measurements had to be deleted; leaving 81% of the desired measurements for analysis.

**Day-to-day variation, inter-investigator variation and left versus right side of the tongue (Table 2).** A significant day-to-day variation in both the PBR (0.012 μm) and the RBC filling (0.003%) was observed for means of both 3 and 5 measurements (p = 0.022 and 0.007, respectively). PBR (SD) means of 3 and 5 measurements was 2.078 (0.264)μm and 2.049 (0.239)μm for day 1 and 1.967 (0.224)μm and 1.911(0.191)μm for day 2, respectively. No significant variation was observed between investigators where PBR (SD) means of 3 was 2.078 (0.264)μm for investigator 1 and 2.068 (0.256)μm for investigator 2 (p = 0.86) and PBR

Table 1. Population characteristics.

| Variable | Participants randomized | Number of participants |
|---|---|---|
| Age (years) | 24.9 (6.1) | 42 |
| Male sex, % | 52 | 42 |
| Smoking duration (pack years†) | 3.3 [1.5–5.0] | 42 |
| Height (m) | 1.75 (0.09) | 42 |
| Weight (kg) | 72.5 (16.4) | 42 |
| Body mass index (kg/m$^2$) | 23.6 (4.8) | 42 |
| Systolic blood pressure (mmHg) | 114 (11) | 42 |
| Diastolic blood pressure (mmHg) | 71 (9) | 42 |
| HbA$_{1c}$ (mmol/mol) | 33 (2.4) | 42 |
| HbA$_{1c}$ (%) | 5.4 (0.36) | 42 |
| Total cholesterol (mmol/L) | 4.2 (0.85) | 42 |
| HDL cholesterol (mmol/L) | 1.4 (0.35) | 42 |
| LDL cholesterol (mmol/L) | 2.3 (0.70) | 42 |
| Triglyceride (mmol/L) | 1.15 (0.61) | 42 |
| PBR means of 3 (μm) | 2.002 [1.807–2.161] | 35 |
| RBC filling means of 3 (μm) | 0.712 [0.656–0.742] | 35 |
| PBR means of 5 (μm) | 1.970 [1.843–2.109] | 30 |
| RBC filling means of 5 (μm) | 0.719 [0.696–0.739] | 30 |

Abbreviations: HDL: High density lipoprotein; LDL: Low density lipoprotein; N: number of participants; PBR:
Perfused Boundary Region; RBC: Red blood cell.
† Number of years smoked x cigarettes a year/20.
Data are percentage (%), mean (SD) or median [IQR].

means of 5 was 2.049 (0.239)μm for investigator 1 and 2.066 (0.227)μm for investigator 2
(p = 0.76). Between left and right side of the tongue no variation in PBR was observed; PBR
means of 3 was 2.066 (0.263)μm for the left side and 2.130 (0.286)μm for the right side
(p = 0.12) and PBR means of 5 was 2.005 (0.221)μm for the left side and 2.119 (0.260)μm for
the right side (p = 0.054). For RBC filling a significant difference was observed for the means
of both 3 and 5 measurements on the left side of the tongue (0.691 (0.062)% and 0.710
(0.049)%, respectively) compared to the right side (0.661 (0.075)% and 0.679 (0.057)%);
p = 0.004 and 0.009, respectively.

Bland-Altman plots were constructed for the above variations (S1–S3 Figs). In general,
there was an improved reproducibility with increasing number of measurements (from 1,
means of 3 to means of 5). The underlying mean value did not influence the mean difference
in any of the plots.

The intraclass correlations were as follows: PBR = 0.30 (confidence interval (CI) 0.20 to
0.40), PBR means of 3 = 0.50 (CI 0.38 to 0.62), PBR means of 5 = 0.58 (CI 0.46 to 0.71), RBC
filling = 0.45 (CI 0.34 to 0.56), RBC filling means of 3 = 0.65 (0.54 to 0.76), and RBC filling
means of 5 = 0.71 (CI 0.61 to 0.82). When correcting for increasing number of measurements
averaged for 3 and 5 measurements respectively: PBR means of 3 = 0.75 (CI 0.66 to 0.84), PBR
means of 5 = 0.87 (CI 0.82 to 0.93), RBC filling means of 3 = 0.85 (0.79 to 0.91), and RBC filling
means of 5 = 0.93 (CI 0.89 to 0.96).

The correlation between the haematocrit and the capillary measures at visit 1 was not significant (p≥0.05).

**Table 2. Variations between days, investigators and side-of-tongue based on single, or mean of three and five measurements.**

| Parameters | Measurement 1 Median [IQR, N] | Measurement 2 Median [IQR, N] | Difference in Parameters (95% CI) | P-value |
|---|---|---|---|---|
| Day-to-day variation | | | | |
| PBR (µm) | 2.044 [1.749–2.350, 40] | 1.918 [1.776–2.322, 34] | -0.001 (-0.015 to 0.013) | 0.89 |
| PBR means of 3 (µm) | 2.112 [1.842–2.257, 359 | 1.972 [1.827–2.056, 31] | -0.010 (-0.019 to -0.002) | **0.022** |
| PBR means of 5 (µm) | 2.070 [0.1.855–2.201, 29] | 1.901 [1.762–2.033, 26] | -0.012 (-0.020 to -0.003) | **0.007** |
| RBC Filling (%) | 0.682 [0.645–0.746, 40) | 0.721 [0.671–0.766, 34] | 0.002 (-0.001 to 0.004 | 0.26 |
| RBC Filling means of 3 (%) | 0.686 [0.637–0.742, 35] | 0.737 [0.689–0.752, 31] | 0.003 (0.001 to 0.005) | **0.008** |
| RBC Filling means of 5 (%) | 0.703 [0.676–0.742, 29] | 0.738 [0.705–0.759, 26] | 0.003 (0.001 to 0.005) | **0.005** |
| Inter investigator variation | | | | |
| PBR (µm) | 2.044 [1.749–2.350, 40] | 2.053 [1.838–2.316, 39] | 0.032 (-0.102 to 0.166) | 0.63 |
| PBR means of 3 (µm) | 2.112 [1.842–2.257, 35] | 2.071 [1.851–2.165, 36] | -0.010 (-0.125 to 0.105) | 0.86 |
| PBR means of 5 (µm) | 2.070 [1.855–2.201, 29] | 2.017 [1.866–2.178, 32] | 0.0154 (-0.089 to 0.120) | 0.76 |
| RBC Filling (%) | 0.682 [0.645–0.746, 40] | 0.691 [0.642–0.731, 39] | -0.002 (-0.026 to 0.021) | 0.83 |
| RBC Filling means of 3 (%) | 0.686 [0.637–0.742, 35] | 0.702 [0.641–0.726, 36] | -0.002 (-0.024 to 0.020) | 0.87 |
| RBC Filling means of 5 (%) | 0.703 [0.676–0.742, 30] | 0.705 [0.649–0.731, 32] | -0.008 (-0.027 to 0.011) | 0.41 |
| Side-of-tongue variation | | | | |
| PBR (µm) | 2.146 [1.846–2.358, 39] | 2.053 [1.779–2.342, 39] | -0.044 (-0.173 to 0.085) | 0.49 |
| PBR means of 3 (µm) | 2.086 [1.869–2.222, 34] | 2.145 [1.951–2.290, 36] | 0.073 (-0.019 to 0.164) | 0.12 |
| PBR means of 5(µm) | 2.001 [1.831–2.184, 27] | 2.115 [1.993–2.332, 31] | 0.091 (-0.002 to 0.183) | 0.054 |
| RBC Filling (%) | 0.684 [0.622–0.722, 39] | 0.668 [0.633–0.736, 39] | 0.006 (-0.020 to 0.031) | 0.65 |
| RBC Filling means of 3 (%) | 0.711 [0.665–0.737, 34] | 0.656 [0.616–0.718, 36] | -0.027 (-0.045 to -0.009) | **0.004** |
| RBC Filling means of 5 (%) | 0.716 [0.676–0.752, 27] | 0.682 [0.648–0.724, 31] | -0.025 (-0.424 to -0.007) | **0.009** |

Results are reported as median (interquartile range), change in parameters (confidence limits). Linear mixed-effects model was used as statistical analysis to account for the correlation of repeated measurements within participants. Measurements: Day-to-day: 1 = first day, 2 = second day; investigator: 1 = investigator 1, 2 = investigator 2; Side-of-tongue: 1 = left side, 2 = right side;

IQR: Interquartile range; N: number; PBR: Perfused Boundary Region; RBC: Red blood cell.

**No intervention (Table 3).** The baseline PBR means (SD) of 3 and 5 measurements was 2.088 (0.283)µm and 2.054 (0.265)µm, respectively and only minor changes over the 180 minutes were observed; the RBC filling remained unchanged.

**Smoking a cigarette (Table 4).** PBR for means of 3 and 5 measurements starting at 2.067 (0.248)µm and 2.043 (0.208)µm at baseline, respectively, increased significantly after 40 and 60 minutes both for means of 3 and 5 measurements compared to baseline. Moreover, changes for PBR were seen for means of 3 measurements at time 120 and 180 minutes and for means of 5 measurements at time 90 and 180 minutes. Highest numerical change was an increase of 0.042 and 0.041µm at 40 minutes both for means of 3 and 5 measurements. For RBC filling, a significant decrease was seen at time 40 and 60 minutes for means of 3 measurements and at time 60 and 180 minutes for means of 5 measurements.

**High-calorie meal intake (Table 5).** Starting at baseline a PBR for means of 3 and 5 measurements of 1.935 (0.191)µm and 1.867 (0.182)µm a significant increase in PBR and decrease in RBC filling for both means of 3 and 5 measurements were seen immediately after food intake. This was also the timepoint where the changes in both parameters were largest, PBR changed being 0.116µm and 0.183µm for means of 3 and 5 measurements, respectively. The changes lasted until the end of the measuring period at 180 minutes though it was attenuated, except from RBC filling, means of 3 measurements at 30 minutes which was not significantly changed.

**Table 3. Results of no intervention before and at times 0–180 min.**

| | Before/ baseline (SD) | 0 min | 20 min | 40 min | 60 min | 90 min | 120 min | 150 min | 180 min |
|---|---|---|---|---|---|---|---|---|---|
| PBR means of 3 (μm) | 2.088 (0.283) N = 19 | 0.029 (-0.009 to 0.066) N = 17 | 0.002 (-0.029 to 0.03169) N = 18 | 0.022 (-0.002 to 0.046) N = 19 | 0.012 (-0.010 to 0.034) N = 18 | 0.009 (-0.006 to 0.024) N = 18 | 0.007 (-0.006 to 0.014) N = 18 | 0.013 (-0.003 to 0.029) N = 17 | 0.005 (-0.008 to 0.018) N = 18 |
| PBR means of 5 (μm) | 2.054 (0.265) N = 17 | **0.034 (0.006 to 0.062)\* N = 16** | 0.015 (-0.010 to 0.039) N = 16 | 0.021 (-0.001 to 0.044) N = 16 | 0.194 (-0.002 to 0.041) N = 17 | 0.013 (-0.00004 to 0.026) N = 16 | **0.010 (0.002 to 0.017)\* N = 17** | **0.017 (0.001 to 0.033)\* N = 16** | 0.007 (-0.005 to 0.018) N = 17 |
| RBC Filling means of 3 (%) | 0.681 (0.067) N = 19 | -0.003 (-0.013 to 0.006) N = 17 | 0.001 (-0.004 to 0.007) N = 18 | -0.002 (-0.006 to 0.003) N = 19 | 0.001 (-0.004 to 0.006) N = 18 | -0.0001 (-0.004 to 0.003) N = 18 | 0.001 (-0.002 to 0.003) N = 18 | -0.001 (-0.005 to 0.003) N = 17 | 0.001 (-0.002 to 0.004) N = 18 |
| RBC Filling means of 5 (%) | 0.697 (0.056) N = 17 | -0.004 (-0.012 to 0.003) N = 16 | -0.002 (- 0.006 to 0.003) N = 16 | -0.003 (-0.007 to 0.002) N = 16 | -0.002 (-0.007 to 0.002) N = 17 | -0.001 (-0.005 to 0.003) N = 16 | -0.001 (0.003 to 0.001) N = 17 | -0.002 (-0.006 to 0.001) N = 16 | -0.001 (-0.003 to 0.002) N = 17 |

Before = the starting value before intervention, mean (standard deviation). 0–180 min shows change over time with no intervention investigating means of 3 and 5 measurements of PBR and RBC filling respectively, compared to means of 3 and 5 measurements before/ at baseline (confidence limits, p-value); N = number of subjects included in the analysis.

\*p<0.05. Linear mixed-effects model was used as statistical analysis to account for the correlation of repeated measurements within participants. PBR: Perfused Boundary Region; RBC: Red blood cell.

**Coffee consumption (Table 6).** Starting at a PBR for means of 3 and 5 measurements of 2.011 (0.264)μm and 1.980 (0.192)μm at baseline, we also observed a significant increase in PBR and a decrease in RBC filling for both means of 3 and 5 measurements already at time 0. Again, the largest change was at time 0 for PBR for means of 3 and 5 measurements being

**Table 4. Results of smoking a cigarette before and at times 0–180 min.**

| | Before/ baseline (SD) | 0 min | 20 min | 40 min | 60 min | 90 min | 120 min | 150 min | 180 min |
|---|---|---|---|---|---|---|---|---|---|
| PBR means of 3 (μm) | 2.067 (0.248) N = 16 | 0.033 (-0.019 to 0.084) N = 12 | 0.028 (-0.009 to 0.065) N = 16 | **0.042 (0.011 to 0.073)\* N = 16** | **0.034 (0.017 to 0.051) ¤ N = 16** | 0.017 (-0.005 to 0.039) N = 16 | **0.019 (0.002 to 0.037)\* N = 16** | 0.015 (-0.0003 to 0.030) N = 16 | **0.014 (0.001 to 0.027)\* N = 16** |
| PBR means of 5 (μm) | 2.043 (0.208) N = 12 | 0.044 (-0.00005 to 0.087) N = 11 | 0.034 (-0.0003 to 0.068) N = 13 | **0.041 (0.003 to 0.079)\* N = 12** | **0.034 (0.018 to 0.0510)¤ N = 13** | **0.025 (0.001 to 0.048)\* N = 14** | 0.018 (-0.003 to 0.380) N = 13 | 0.015 (-0.002 to 0.032) N = 13 | **0.015 (0.002 to 0.029)\* N = 10** |
| RBC Filling means of 3 (%) | 0.686 (0.066) N = 16 | -0.007 (-0.017 to 0.004) N = 11 | -0.003 (-0.011 to 0.005) N = 16 | **-0.007 (-0.014 to -0.0003)\* N = 16** | **-0.007 (-0.010 to -0.003)# N = 16** | -0.002 (-0.007 to 0.003) N = 16 | -0.002 (-0.006 to 0.002) N = 16 | -0.002 (-0.004 to 0.001) N = 16 | -0.002 (-0.004 to 0.001) N = 16 |
| RBC Filling means of 5 (%) | 0.0704 (0.056) N = 12 | -0.010 (-0.020 to 0.001) N = 12 | -0.006 (-0.015 to 0.003) N = 13 | -0.009 (-0.018 to 0.001) N = 12 | **-0.007 (-0.012 to -0.002)# N = 13** | -0.005 (-0.011 to 0.002) N = 14 | -0.003 (-0.008 to 0.001) N = 13 | -0.003 (-0.006 to 0.0003) N = 13 | **-0.003 (-0.005 to -0.0003)\* N = 10** |

Before = the starting value before intervention, mean (standard deviation). 0–180 min shows change over time after smoking a cigarette investigating means of 3 and 5 measurements of PBR and RBCfilling respectively, compared to means of 3 and 5 measurements before/ at baseline (confidence limits, p-value); N = number of subjects included in the analysis.

\*p<0.05,

#p<0.01,

¤p<0.001. Linear mixed-effects model was used as statistical analysis to account for the correlation of repeated measurements within participants. PBR: Perfused Boundary Region; RBC: Red blood cell.

**Table 5. Results of a high calorie meal before and at times 0–180 min.**

| | Before/ baseline (SD) | 0 min | 30 min | 60 min | 90 min | 120 min | 150 min | 180 min |
|---|---|---|---|---|---|---|---|---|
| PBR means of 3 (µm) | 1.935 (0.191) N = 18 | **0.116 (0.055 to 0.177)¤ N = 19** | **0.063 (0.022 to 0.104)# N = 19** | **0.075 (0.036 to 0.114)¤ N = 18** | **0.057 (0.035 to 0.079)¤ N = 18** | **0.048 (0.028 to 0.068)¤ N = 18** | **0.039 (0.023 to 0.054)¤ N = 19** | **0.032 (0.019 to 0.045)¤ N = 19** |
| PBR means of 5 (µm) | 1.867 (0.182) N = 16 | **0.183 (0.116 to 0.250)¤ N = 17** | **0.073 (0.029 to 0.116)# N = 17** | **0.099 (0.069 to 0.129)¤ N = 17** | **0.064 (0.047 to 0.082)¤ N = 17** | **0.062 (0.040 to 0.083)¤ N = 17** | **0.050 (0.034 to 0.065)¤ N = 17** | **0.033 (0.021 to 0.045)¤ N = 17** |
| RBC Filling means of 3 (%) | 0.723 (0.059) N = 18 | **-0.027 (-0.047 to -0.006)* N = 19** | -0.012 (-0.026 to 0.002) N = 19 | **-0.015 (-0.023 to -0.006)# N = 18** | **-0.011 (-0.016 to -0.006)¤ N = 18** | **-0.009 (-0.014 to -0.004)# N = 18** | **-0.007 (-0.012 to -0.002)# N = 19** | **-0.005 (-0.0081 to -0.002)# N = 19** |
| RBC Filling means of 5 (%) | 0.739 (0.050) N = 16 | **-0.035 (-0.054 to -0.016)# N = 17** | **-0.012 (0.023 to -0.001)* N = 17** | **-0.018 (-0.025 to -0.010)¤ N = 17** | **-0.012 (0.016 to -0.007)¤ N = 17** | **0.012 (-0.015 to -0.006)¤ N = 17** | **-0.009 (-0.013 to -0.005)¤ N = 17** | **-0.005 (-0.008 to -0.002)# N = 17** |

Before = the starting value before intervention, mean (standard deviation). 0–180 min shows change over time after eating a high calorie meal investigating means of 3 and 5 measurements of PBR and RBCfilling respectively, compared to means of 3 and 5 measurements before/ at baseline (confidence limits, p-value); N = number of subjects included in the analysis.

*p<0.05,

#p<0.01,

¤p<0.001. Linear mixed-effects model was used as statistical analysis to account for the correlation of repeated measurements within participants. PBR: Perfused Boundary Region; RBC: Red blood cell.

0.141µm and 0.160µm, respectively and attenuated thereafter. For PBR both with means of 3 and 5 measurements the significant changes lasted for 180 minutes except at time 120 and 150 minutes for means of 3 and at 60 and 90 minutes for means of 5 measurements. For RBC filling means of 3 measurements the change lasted till 90 minutes and for RBC filling means of 5 measurements until 180 minutes though not at time 60 minutes.

**Table 6. Results of a drinking a cup of coffee before and at times 0–180 min.**

| | Before/ baseline (SD) | 0 min | 30 min | 60 min | 90 min | 120 min | 150 min | 180 min |
|---|---|---|---|---|---|---|---|---|
| PBR means of 3 (µm) | 2.011 (0.264) N = 13 | **0.141 (0.018 to 0.265)* N = 12** | **0.106 (0.024 to 0.188)* N = 12** | **0.063 (0.009 to 0.117)* N = 12** | **0.060 (0.018 to 0.101)# N = 12** | 0.018 (-0.006 to 0.042) N = 12 | 0.019 (-0.006 to 0.045) N = 12 | **0.018 (0.0002 to 0.036)* N = 12** |
| PBR means of 5 (µm) | 1.980 (0.192) N = 10 | **0.160 (0.024 to 0.230)* N = 10** | **0.107 (0.024 to 0.189)* N = 10** | 0.060 (-0.003 to 0.124) N = 10 | 0.052 (-0.007 to 0.110) N = 10 | **0.023 (0.002 to 0.045)* N = 10** | **0.028 (0.008 to 0.048)* N = 10** | **0.026 (0.006 to 0.046)* N = 10** |
| RBC Filling means of 3 (%) | 0.708 (0.064) N = 13 | **-0.033 (-0.059 to 0.007)* N = 12** | **-0.019 (0.032 to 0.005)* N = 12** | **-0.014 (-0.023 to -0.005)# N = 12** | **-0.011 (-0.018 to 0.005)# N = 12** | -0.004 (-0.010 to 0.002) N = 12 | -0.003 (-0.009 to 0.002) N = 12 | -0.003 (-0.007 to 0.001) N = 12 |
| RBC Filling means of 5 (%) | 0.726 (0.027) N = 10 | **-0.043 (-0.069 to -0.018)# N = 10** | **-0.019 (-0.030 to -0.008)# N = 10** | -0.013 (-0.026 to 0.00005) N = 10 | **-0.011 (-0.019 to -0.003)* N = 10** | **-0.006 (-0.008 to -0.003)¤ N = 10** | **-0.007 (-0.011 to -0.002)# N = 10** | **-0.006 (-0.009 to -0.002)# N = 10** |

Before = the starting value before intervention, mean (standard deviation). 0–180 min shows change over time after drinking coffee investigating means of 3 and 5 measurements of PBR and RBCfilling respectively, compared to means of 3 and 5 measurements before/ at baseline (confidence limits, p-value); N = number of subjects included in the analysis.

*p<0.05,

#p<0.01,

¤p<0.001. Linear mixed-effects model was used as statistical analysis to account for the correlation of repeated measurements within participants. PBR: Perfused Boundary Region; RBC: Red blood cell.

## Discussion

In this study including 42 young healthy subjects, measurements with the GlycoCheck system showed no differences between measurements of PBR and RBC filling performed by two different investigators. Likewise, PBR and RBC filling had good inter-observer reproducibility as judged by the Bland-Altman plots. For PBR no difference between the side of tongue was found; however, for RBC filling a higher value was observed on the right side of the tongue. We found minor differences in day-to-day measurements. The intra-class correlations showed only a modest reproducibility but was highly improved with higher numbers of measurements. The ICC was higher when the calculation included the number of measurements. Effects of intervention with both a cigarette, a high calorie meal and a cup of coffee were observed. Measurements of PBR and RBC filling over time for the same period but without any intervention, resulted in a few significant small changes, but with not no clear pattern.

The GlycoCheck system is an easy, non-invasive method for measurement of the glycocalyx offering potential important clinical information. The importance of the glycocalyx for several processes associated with the vessel wall is increasingly being acknowledged, including coagulation, lipid handling, immune response and microvascular perfusion [3, 8–10]. Damage to the glycocalyx has been considered an important phenomenon in relation to endothelial dysfunction in diverse conditions like atherosclerosis and sepsis, and has been considered an early event in progression of diabetic kidney disease, where damage to the glycocalyx in the glomeruli increases the degree of albuminuria as demonstrated in a mouse model [12]. Thus, a damaged glycocalyx measured locally could be viewed as an expression of generalised microvascular damage and possibly a precursor of complications as seen e.g. in diabetes with retinopathy, nephropathy, etc. [29] which emphasises the importance of easy, non-invasive measurement.

Reproducibility of the GlycoCheck system has been examined by others. Investigating 70 patients admitted to the emergency room and the intensive care unit Rovas et al. found an excellent intra-observer reproducibility (ICC at 0.77 (CI 0.52–0.89) for PBR and 0.88 (CI 0.74–0.94) for RBC filling) and when comparing measurements performed by nurses and doctors they found a high reproducibility (ICC at 0.75 (CI 0.52–0.87) [21]. Valerio et al. combined two studies of 50 participants of multi-ethnic origin where two measurements were performed one minute apart and 21 healthy controls with 6 measurements separated by 3 minutes. They found an ICC (CI) of 0.33 (0.08–0.56) for PBR and 0.51 (0.27–0.69) for RBC filling [19]. These results are similar to ours for single measurements (ICC at 0.30 (CI 0.20 to 0.40) for PBR and 0.45 (CI 0.34 to 0.56) for RBC filling. Valerio et al. did not perform series of measurements for mean calculations [19]. Weissgerber et al. investigated 13 pregnant women in a day-to-day reliability study. They found that reliability between individual measurements was low with one measurement however with an average of three they primarily found a moderate (ICC 0.50–0.75) to excellent (>0.90) day-to-day reliability though only in a subset of vessel sizes ranging from 9 to 18 μm and only excellent for 12 μm vessels [22]. Also investigating ICU patients Bol et al. included 60 participants and in a subset of 20 participants inter-observer reproducibility was investigated. They found an increasing reproducibility with increasing number of measurements reaching ICCs of 0.80 [IQR 0.68–0.88] and 0.66 [IQR 0.37–0.83] for PBR and 0.90 [IQR 0.84–0.94] and 0.79 [IQR 0.61–0.89] for RBC filling for the two investigators respectively. They found no systematic difference between observers [20]. It must be stressed, that only one of the described studies state the form of ICC used. Different ICC forms can lead to significant differences, why comparisons should be performed with caution [25].

The ICC in our study improved with increasing numbers of measurements, reaching 0.54 (CI 0.38 to 0.62) and 0.65 (CI 0.54 to 0.76) for means of 3 measurements for PBR and RBC

filling respectively and 0.59 (CI 0.46 to 0.71) and 0.71 (CI 0.61 to 0.82) for means of 5 measurements for PBR and RBC filling respectively. The ICC increased markedly when the number of measurements was included in the calculation reaching 0.75 (CI 0.66 to 0.84) and 0.85 (0.79 to 0.91) for means of 3 measurements for PBR and RBC filling, respectively and 0.87 (CI 0.82 to 0.93) and 0.93 (CI 0.89 to 0.96) for means of 5 measurements for PBR and RBC filling, respectively. Thus, a moderately acceptable level was reached with 5 measurements however only with CI above the threshold for RBC filling with results greatly improving when number of measurements was included in the statistical analysis. With means of both 3 and 5 measurements we were able to detect the effect of interventions. This confirms the reliability of the measurements but stresses the importance of several measurements being performed when using the GlycoCheck system where at least three measurements would be advisable.

Change in PBR of 0.2 μm was suggested to be of clinical relevance based on a study comparing 50 people with premature coronary artery disease to the same amount of both family members and healthy controls [23]. From the achieved standard deviation, a clinical study of an intervention would require only 20 subjects in each arm to demonstrate such a clinically relevant difference.

In a community-based study investigating 915 subjects, Lee et al. found mean PBR (SD) of 2.14 (0.25)μm, only slightly higher than ours at 1.995 (0.179)μm most likely corresponding to the older age of 56.1 (SD = 6.0) of their group. Their results were with a wide range of variability (range 1.43 to 2.86 μm) however they were able to find small differences between groups of 0.09μm [14]. This corresponds well to our finding of a low ICC and thus a weak correlation between measurements in the same individual yet finding small significant changes with exposures. Donati et al. found a coefficient of variation for PBR between three consecutive measurements in 7 healthy subjects to be 0.05±0.05, thus small but with a large standard deviation [30]. They hence find the measurements to be with reasonable reproducibility however the results being uncertain. With an average value of 2.46μm this would correspond to a difference of 0.12μm but up to 0.37μm, which is a larger difference than our findings of up to 0.264 μm for means of three measurements and smaller for means of five measurements.

Measurements with the GlycoCheck devise are assumed to reflect a general status of the systemic glycocalyx [13]. As all three performed interventions are oral it should be noted that the observed effects of the interventions might be due to a transient local effect and not systemic damage to the glycocalyx.

Overall, smoking negatively influenced PBR and RBC filling between 40 and 180 minutes after smoking. The effect might however last longer than our study period of 180 minutes.

In the above-mentioned study by Valerio et al. in 21 healthy non-smokers, the impact of smoking was assessed with a measurement performed at baseline and repeated after the participant smoked two cigarettes. The effect of smoking was only evaluated after 5 minutes and like us they found no effect of smoking at this timepoint. Furthermore, there was no information on the interval between smoking of the two cigarettes [19]. Had they performed consecutive measurements like us they might have found an effect. Based on our study showing an acute effect of smoking, we recommend that subjects should refrain from smoking at least three hours prior to the GlycoCheck measurements.

Consumption of a high calorie meal had an immediate negative effect on both PBR and RBC filling lasting at least up till 180 minutes but attenuating. High calorie meal intake is assumed to be associated with and negatively influencing endothelial function. In the study by Valerio et al. the impact of a meal was investigated in 50 participants. Measurements were performed only at one point one hour after intake of the meal finding no impact on neither PBR nor RBC filling [19]. Their standardised meal was 390 kcal equalling 1632 kJ, considerably less than the 4185 kJ in our study. Furthermore, we found the effect to be largest immediately after

the intake of the meal tapering off afterwards, which indicates that an effect of a smaller meal might have subsided after one hour. Kackov et al. investigated the effect of a high calorie meal on markers associated with endothelial dysfunction. They demonstrated a significant increase in total antioxidant status and intercellular adhesion molecule-1; however, only measurements at baseline and 3 hours after the meal were performed [31]. This is in accordance with our findings, indicating a decrease in glycocalyx thickness after a high calorie meal, using a different measuring technique. It must be stressed, that this comparison only holds true if the effect measured with the GlycoCheck camera is systemic and not local, however still suggesting subjects to be fasting for at least 180 minutes before measurement.

Drinking coffee had an immediate negative effect on GlycoCheck measurements lasting up till 180 minutes. This is surprising as coffee consumption has been demonstrated to have an inverse relationship to both all-cause mortality and death of heart disease [32]. Others have however demonstrated a negative acute effect of coffee consumption though on other markers reflecting endothelial function. Buscemi et al. investigated the effect of ingestion of caffeinated coffee compared to non-caffeinated coffee on brachial artery flow-mediated dilation (FMD). This in 20 healthy individuals demonstrating a decrease in FMD reflecting a negative effect [33]. Again, one can only compare when assuming a systemic effect, yet our findings indicate that subjects should refrain from coffee consumption at least for 3 hours prior to the investigation.

The possibility that the investigated effects of the three abovementioned interventions are local and not systemic does not change our recommendations to refrain from intake of either of the three at least three hours prior to investigations with the GlycoCheck system.

Damage to glycocalyx has shown to be associated with diseases such as the presence of atherosclerosis [18, 23]. Using the GlycoCheck system, Amraoui et al. could however not demonstrate this association in a small study investigating the difference in glycocalyx between healthy people, people with low or high risk of cardiovascular disease and people with manifest cardiovascular disease [24]. Using the GlycoCheck system in a cross-sectional study of a community-based cohort of 6169 individuals Valerio et al. also found no association between PBR and cardiovascular disease. They did however demonstrate an association between a damaged glycocalyx and diabetes (diabetes being defined as fasting plasma glucose $\geq$ 7 mmol/l, or the use of glucose-lowering medication) [34] an association also demonstrated by others in smaller studies [13, 35]. Another cross-sectional community-based study of 915 participants demonstrated an association between PBR and RBC filling–both measured with the GlycoCheck system. This underlines the importance of the glycocalyx in various disease processes as impaired microvascular perfusion can result in damage to several organs. The authors suggested that the glycocalyx assists in elongating the red blood cells and aligns them for easier passage into the capillaries [14]. In general, there are few studies on SDF coupled to this new software and more studies are needed.

A benefit with the GlycoCheck system is that it is non-invasive, without any discomfort for the patient and easy to use. Furthermore, the investigators were easily trained according to the manufacturer's guidance. Limitations in this study were technical issues with the camera. This resulted in exclusion of 19% of the measurements. Using mixed model analysis accounted for the missing data in the statistical analysis. As data were missing due to a defect equipment (technical difficulties) and not related to clinical characteristics there was no selection bias and data were missing at random. Other studies have experienced similar exclusion rates [34, 36]. As this is a study on young healthy individuals, a limitation is the uncertainty of an extrapolation of the accuracy of the measurements to groups which are different with regards to age or disease status. We asked the participants to be fasting before both visits however we could not control their compliance to this, which is a limitation. We cannot exclude that changes in

hydration status could impact the measures, but at baseline there was no association with hae-matocrit and PBR and RBC filling.

In conclusion, we found measurements of glycocalyx and microvascular perfusion with the GlycoCheck system to be with a moderate reproducibility, but markedly improved with multiple measurements, and measures affected by examination conditions. We therefore recommend that measurements should be performed in the fasting state and without coffee consumption and smoking for at least 180 minutes prior to the investigation. However, the GlycoCheck system needs further validation before implementation as a standard measure of risk prediction.

## Supporting information

**S1 Fig. Bland Altman plots on day-to-day variation of perfused boundary region (PBR).**
(TIF)

**S2 Fig. Bland Altman plots on variation between investigators of perfused boundary region (PBR).**
(TIF)

**S3 Fig. Bland Altman plots on variation between side of toungue of perfused boundary region (PBR).**
(TIF)

## Acknowledgments

We thank all the participants and the lab technicians Berit R. Jensen, Tina R. Juhl, Jessie A. Hermann and Anne G Lundgaard, employees at Steno Diabetes Center Copenhagen.

## Author Contributions

**Conceptualization:** Mie Klessen Eickhoff, Signe Abitz Winther, Tine Willum Hansen, Frederik Persson, Peter Rossing, Marie Frimodt-Møller.

**Data curation:** Mie Klessen Eickhoff, Lars Jorge Diaz.

**Formal analysis:** Mie Klessen Eickhoff, Lars Jorge Diaz.

**Funding acquisition:** Peter Rossing.

**Investigation:** Mie Klessen Eickhoff, Signe Abitz Winther, Tine Willum Hansen, Frederik Persson, Peter Rossing, Marie Frimodt-Møller.

**Methodology:** Mie Klessen Eickhoff, Signe Abitz Winther, Tine Willum Hansen, Frederik Persson, Peter Rossing, Marie Frimodt-Møller.

**Project administration:** Mie Klessen Eickhoff, Frederik Persson, Peter Rossing, Marie Frimodt-Møller.

**Resources:** Peter Rossing.

**Software:** Mie Klessen Eickhoff, Lars Jorge Diaz.

**Supervision:** Peter Rossing, Marie Frimodt-Møller.

**Validation:** Mie Klessen Eickhoff, Peter Rossing, Marie Frimodt-Møller.

**Visualization:** Mie Klessen Eickhoff, Peter Rossing.

**Writing – original draft:** Mie Klessen Eickhoff.

**Writing – review & editing:** Mie Klessen Eickhoff, Signe Abitz Winther, Tine Willum Hansen, Lars Jorge Diaz, Frederik Persson, Peter Rossing, Marie Frimodt-Møller.

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
