## [Decision Letter · Decision Letter 0]

13 May 2020

PONE-D-20-03635

Assessment of the Sublingual Microcirculation with the GlycoCheck System:

Reproducibility and Examination Conditions

PLOS ONE

Dear Dr. Frimodt-Møller,

Thank you for submitting your manuscript to PLOS ONE. After careful consideration, we feel that it has merit but does not fully meet PLOS ONE’s publication criteria as it currently stands. Therefore, we invite you to submit a revised version of the manuscript that addresses the points raised during the review process.

We would appreciate receiving your revised manuscript by Jun 27 2020 11:59PM. To enhance the reproducibility of your results, we recommend that if applicable you deposit your laboratory protocols in protocols.io, where a protocol can be assigned its own identifier (DOI) such that it can be cited independently in the future. For instructions see: http://journals.plos.org/plosone/s/submission-guidelines#loc-laboratory-protocols

We look forward to receiving your revised manuscript.

Kind regards,

Mateusz K. Holda, MD, PhD

Academic Editor

PLOS ONE

Journal Requirements:

The study was approved by the Copenhagen local ethics committee and was conducted according to the declaration of Helsinki. Protocol nr. H-15010254

3. In the online submission form please clarify whether any competing interests related to the manufacturer of the GlycoCheck System exist. Thank you for your attention to this request.

'I have read the journal's policy and the authors of this manuscript have the following

competing interests: MKE has served as educator for Astra Zeneca (all honoraria’s to institution). FP has served as consultant, on advisory boards or as educator for Astra Zeneca,

Novo Nordisk, Sanofi, Mundipharma, MSD, Boehringer Ingelheim, Novartis, Amgen

and has received research grants to institution from Novo Nordisk, Amgen and Astra

Zeneca.

PR has served as consultant, on advisory boards or as educator for Astra Zeneca,

Astellas, AbbVie, Novo Nordisk, Boehringer Ingelheim, Eli Lilly, Merck, Bayer (all

honoraria’s to institution), has shares in Novo Nordisk and has received research

grants to institution from Novo Nordisk and Astra Zeneca.'

Additional Editor Comments (if provided):

Reviewers' comments:

Reviewer's Responses to Questions

**Comments to the Author**

1. Is the manuscript technically sound, and do the data support the conclusions?

Reviewer #1: Yes

Reviewer #2: Partly

2. Has the statistical analysis been performed appropriately and rigorously? 

Reviewer #1: Yes

Reviewer #2: No

3. Have the authors made all data underlying the findings in their manuscript fully available?

Reviewer #1: Yes

Reviewer #2: Yes

4. Is the manuscript presented in an intelligible fashion and written in standard English?

Reviewer #1: Yes

Reviewer #2: Yes

5. Review Comments to the Author

Reviewer #1: The authors are presenting an extensive investigation on reproducibility of glycocalyx PBR thickness using the glycocheck system. The authors have evaluated intra- and inter- reader variability as well the effect of examing conditions (smoking, post-prandial status, etc). The study is well described and reads nicely. I only have a few comments which might improve already well written manuscript.

1) Please provide the figure displaying the capture of actual measurements with the device with labeled outer layer and PBR.

2) The authors state that multiple measurements should be performed to achieve reproducible measurements capable of detecting the effect by intervention. There was only slight improvement in ICC when conducting 5 measurements instead of 3. What would be the authors recommendation for standardization in this case?

3) The authors investigated relatively young participants. Would there be an age related effect on the accuracy of the glycocalyx measurements?

4) Whats the effect of hydration on similar capillary measures? Did authors control for this effect?

5) The authors should discuss the potential effect of microvascular disease (T1D) or global vascular disease on these measurements.

Reviewer #2: I would like to thank the editor for the opportunity to review the manuscript entitled „Assessment of the Sublingual Microcirculation with the GlycoCheck System: Reproducibility and Examination Conditions“.

In the last years the endothelial glycocalyx has been shown to play an important role in different diseases and the topic is of great interest. This non-invasive method to measure the glycocalyx dimensions at the bedside can be of great use for further studies and a generally accepted SOP is needed. The language used by the authors is appropriate.

However, I have some major concerns, that I would like to mention:

Introduction:

Page 4, line 64: The reference used is outdated (1966). Indeed, it is a challenge to measure the size of the glycocalyx in extracted tissues. However, nowadays this is possible with the use of different methods (PMID: 17256154 and atomic force microscopy PMID: 24278345, 24727235).

Page 4, line 67-68: I would kindly disagree with the authors at that point. Further studies have been conducted to evaluate the reproducibility of the PBR measurements (31743299, 29444696, 30764695). Moreover, besides reproducibility of the PBR measurements, accuracy of the method has been also already evaluated with the use of atomic force microscopy (31493777, 31340868).

Materials & Methods:

Page 6, line 111-113: Could the fact that Days 1 and 2 were up to 7 days apart be responsible for the variation observed? This could be a limitation of the study. Could the observed PBR difference be the result of a change of the systemic glycocalyx? Median days between Days 1 and 2 should be also reported by the authors.

Moreover, were participants controlled, if they indeed abstained from food, beverages and smoking for eight hours before each visit day? If not, this should be referred in the limitations of the study.

Page 8, line 157-160: The ICC Models used in every occasion should be clearly reported, as their choice could have a tremendous effect on the results. (Trevethan R. Intraclass correlation coefficients: clearing the air, extending some cautions, and making some requests. Health Serv. Outcome Res Methodol. 2016;17(2):127–43.)

Figure 2 – Study design: The study design, as well as how many individuals have been included in each procedure should be better elucidated in the figure.

Moreover, if 758 measurements have been discarded post-hoc, could that have influenced the results? How have the authors dealt with the missing data?

Table 1 & 2: In my opinion, it would be more appropriate to report medians with IQR instead of means with SD. In that way, the influence of possible outliers is being minimized. PBR mean of 3 and PBR mean of 5 should be further reported as means of 3 or 5 measurements.

Results:

Page 9, line 171: Could the authors please indicate what criteria were used to discard the measurements and what technical issues they faced? (e.g. camera malfunction? Food/drink rests in the optic field? Patients‘ discomfort?) Were researchers shown how to avoid pressure artifacts? Did they deliberately try to assess more than one spots in their measurements in order to counterbalance for spatial heterogeneity?

Tables 2 – 6: Could the authors please specify, what statistical test has been used to calculate the p values?

Discussion:

Page 15, line 316-317: Indeed, more studies are needed to further understand this new technic. However, more than “few studies” have been already conducted. A Pubmed search of “perfused boundary region” shows about 50 conducted studies in different settings.

Page 12, line 241-242: As referred above, more studies than reported have been already conducted to evaluate the reproducibility of the PBR measurements under different conditions (e.g. 31743299, 29444696, 30764695).

In my opinion, the results are misinterpreted in the discussion section. The authors postulate that a change in PBR after every intervention (meal, coffee, smoking) reflects a change of the systemic endothelial glycocalyx. However, a local effect would be in that case more likely. For example, warm coffee would probably cause locally vasodilatation and the consumption of a meal would probably affect locally the sublingual mucosa. The effect of coffee could have been maybe evaluated if the effect of warm coffee was compared to the possible effect of warm water.

Regarding the inter-observer reproducibility, that fact that the analysis is being performed in different days (up to 7 days), could play a role for the observed PBR differences. A change of the systemic endothelial glycocalyx during that time cannot also be excluded.

Τhe reported results remain important and help to further standardise the sublingugal glycocalyx measurements and reveal possible interference factors. I would suggest to focus the discussion on the fact, that different factors (e.g. meal, coffee, smoke) seem to intervene significantly with the measurements (locally or systemic cannot be answered by the study design) and that the future researchers need to assess more than one measurements from different positions of the sublingual mucosa to estimate the PBR and RBC filling%.

6. PLOS authors have the option to publish the peer review history of their article (what does this mean?). If published, this will include your full peer review and any attached files.

Reviewer #1: Yes: Michal Schäfer

Reviewer #2: No

---

## [Author Response · Author response to Decision Letter 0]

27 Jun 2020

Dear Sir/ Mam

All responses to specific reviewer and editor comments have been included in the "Response to reviewers".

Thank you for considering this manuscript for publication.

Kind regards,

Marie Frimodt-Møller

---

## [Decision Letter · Decision Letter 1]

3 Aug 2020

PONE-D-20-03635R1

Assessment of the sublingual microcirculation with the GlycoCheck system:

Reproducibility and examination conditions

PLOS ONE

Dear Dr. Frimodt-Møller,

Thank you for submitting your manuscript to PLOS ONE. After careful consideration, we feel that it has merit but does not fully meet PLOS ONE’s publication criteria as it currently stands. Therefore, we invite you to submit a revised version of the manuscript that addresses the points raised during the review process.

We look forward to receiving your revised manuscript.

Kind regards,

Mateusz K. Holda, MD, PhD

Academic Editor

PLOS ONE

Reviewers' comments:

Reviewer's Responses to Questions

**Comments to the Author**

1. If the authors have adequately addressed your comments raised in a previous round of review and you feel that this manuscript is now acceptable for publication, you may indicate that here to bypass the “Comments to the Author” section, enter your conflict of interest statement in the “Confidential to Editor” section, and submit your "Accept" recommendation.

Reviewer #1: All comments have been addressed

Reviewer #2: (No Response)

2. Is the manuscript technically sound, and do the data support the conclusions?

Reviewer #1: Yes

Reviewer #2: Yes

3. Has the statistical analysis been performed appropriately and rigorously? 

Reviewer #1: Yes

Reviewer #2: No

4. Have the authors made all data underlying the findings in their manuscript fully available?

Reviewer #1: Yes

Reviewer #2: Yes

5. Is the manuscript presented in an intelligible fashion and written in standard English?

Reviewer #1: Yes

Reviewer #2: Yes

6. Review Comments to the Author

Reviewer #1: The authors have answered all the raised questions and in my opinion significantly improved the manuscript.

Reviewer #2: I would like to thank the authors for taking my comments into account and revising their manuscript.

However, I still have some concerns:

- ICC (2, k) would be the correct ICC form to address inter-rater reliability, when means of measurements are used, but not to check for test-retest reliability. The latter should be assessed with ICC (3, k). (Please check also 27330520 for more details). In case only two single measurements are being used, then ICC (2,1) or ICC (3,1) should be used. Please address this issue, as indeed different ICC forms can lead to tremendous differences.

- Could the authors please confirm, that PBR means of 5 (e.g. in Table 3) means, that in every timepoint 5 measurements were made and compared with the PBR means of 5 measurements from the baseline?

- Line 261 - 262: Please finish the sentence.

- Rovas et al compared two set of measurements (mean of two measurements vs mean of two further measurements), Valerio et al compared single measurements with each other. Your data, as well as Weissgerber et al and Bol et al confirm the above studies, as they indeed show that the reliability is higher, if means of more than one measurements are being calculated. However, the ICCs of the studies can be only compared directly with cautious, if they are not reported in the original manuscripts. In that case, it might be safer to compare the Bland-Altman plots.

- I would advise the authors, to further tone down regarding the systemic glycocalyx effect of the different interventions and stress that in this study only the local effect was measured. (e.g. Line 305-334).

That does not change the importance of their results, as individuals should refrain from intake before measurements. However, the above passage still gives the false impression, that a direct systemic glycocalyx damage or endothelial dysfunction occurs after coffee consumption or smoking. Although that might indeed be true, it is not supported by the results of this study. To claim the systemic effect of the interventions, either further glycocalyx / endothelial markers should have been measured (e.g. syndecan-1) or the glycocalyx should have been measured on another spot - e.g. ocular.

- Table 4 - what does SIGN mean? In case it means significant, the level of significance should be shown as well.

7. PLOS authors have the option to publish the peer review history of their article (what does this mean?). If published, this will include your full peer review and any attached files.

Reviewer #1: **Yes: **Michal Schäfer

Reviewer #2: No

---

## [Author Response · Author response to Decision Letter 1]

30 Sep 2020

We have added a file entitled Response to reviewers and responded to reviewer comments there.

---

## [Decision Letter · Decision Letter 2]

20 Oct 2020

PONE-D-20-03635R2

Assessment of the sublingual microcirculation with the GlycoCheck system:

Reproducibility and examination conditions

PLOS ONE

Dear Dr. Frimodt-Møller,

Thank you for submitting your manuscript to PLOS ONE. After careful consideration, we feel that it has merit but does not fully meet PLOS ONE’s publication criteria as it currently stands. Therefore, we invite you to submit a revised version of the manuscript that addresses the points raised during the review process.

We look forward to receiving your revised manuscript.

Kind regards,

Mateusz K. Holda, MD, PhD, DSc

Academic Editor

PLOS ONE

Reviewers' comments:

Reviewer's Responses to Questions

**Comments to the Author**

1. If the authors have adequately addressed your comments raised in a previous round of review and you feel that this manuscript is now acceptable for publication, you may indicate that here to bypass the “Comments to the Author” section, enter your conflict of interest statement in the “Confidential to Editor” section, and submit your "Accept" recommendation.

Reviewer #1: All comments have been addressed

Reviewer #2: (No Response)

2. Is the manuscript technically sound, and do the data support the conclusions?

Reviewer #1: Yes

Reviewer #2: Yes

3. Has the statistical analysis been performed appropriately and rigorously? 

Reviewer #1: Yes

Reviewer #2: Yes

4. Have the authors made all data underlying the findings in their manuscript fully available?

Reviewer #1: Yes

Reviewer #2: No

5. Is the manuscript presented in an intelligible fashion and written in standard English?

Reviewer #1: Yes

Reviewer #2: Yes

6. Review Comments to the Author

Reviewer #1: I believe that the authors have provided and extensive revision and answered all minor and major comments.

Reviewer #2: I would like to thank the authors for successfully addressing my comments and concerns.

I only have a minor suggestion:

- Basically, the new ICC analysis shows that multiple measurements (3 to 5) leads to good reproducibility. Therefore, I would suggest integrating in the conclusion the fact that multiple measurements increase the reproducibility of the method dramatically and leads to more robust results.

7. PLOS authors have the option to publish the peer review history of their article (what does this mean?). If published, this will include your full peer review and any attached files.

Reviewer #1: **Yes: **Michal Schafer

Reviewer #2: No

---

## [Author Response · Author response to Decision Letter 2]

27 Nov 2020

We have responded to reviewers in the file entitled "response to reviewers".

---

## [Editor Report · Decision Letter 3]

30 Nov 2020

Assessment of the sublingual microcirculation with the GlycoCheck system:

Reproducibility and examination conditions

PONE-D-20-03635R3

Dear Dr. Frimodt-Møller,

We’re pleased to inform you that your manuscript has been judged scientifically suitable for publication and will be formally accepted for publication once it meets all outstanding technical requirements.

Kind regards,

Mateusz K. Holda, MD, PhD

Academic Editor

PLOS ONE
---

## [Editor Report · Acceptance letter]

7 Dec 2020

PONE-D-20-03635R3 

Assessment of the sublingual microcirculation with the GlycoCheck system:
Reproducibility and examination conditions 

Dear Dr. Frimodt-Møller:

I'm pleased to inform you that your manuscript has been deemed suitable for publication in PLOS ONE. Congratulations! Your manuscript is now with our production department. 

Kind regards, 

on behalf of

Dr. Mateusz K. Holda 

Academic Editor

PLOS ONE